# Disability and pain after lumbar surgery–group-based trajectory analysis

**Konsta Koivunen**[1]*, **Jari Arokoski**[2], **Sara Widbom-Kolhanen**[3], **Katri Pernaa**[4], **Juhani Juhola**[5], **Mikhail Saltychev**[5]

1 Clinical Division, Turku University Hospital and University of Turku, Turku, Finland, 2 Division of Rehabilitation, Department of Internal Medicine and Rehabilitation, Helsinki University Hospital and Helsinki University, Helsinki, Finland, 3 Department of Surgery, Satasairaala Hospital, Pori, Finland, 4 Department of Orthopedics, Turku University Hospital and University of Turku, Turku, Finland, 5 Department of Physical and Rehabilitation Medicine, Turku University Hospital and University of Turku, Turku, Finland

* kokkoi@utu.fi

## Abstract

### Background and aims

Previous studies in lumbar spine surgery have mainly studied functioning and pain by comparing average scores from Patient Reported Outcome Measures (PROMs) at different time points. Less is known about these changes in different subgroups. It is self-evident that, while most patients may demonstrate trajectories of these changes close to the average one, some groups may follow more or less different trends. Also, it is unclear which preoperative factors may affect the probability of being classified into groups with different development trajectories of surgical outcome. The objective of this study was to identify groups exhibiting distinct trajectories within the broader cohort of patients undergoing lumbar spine surgery and to determine whether certain factors may be associated with a probability of being classified into a particular group.

### Methods

This was a register-based study of 1,451 patients undergoing lumbar spine surgery. The group-based trajectory analysis was used separately for leg pain, for back pain, and for functioning. The probability of group membership was calculated based on sex, age, leg and back pain duration before surgery, and obesity.

### Results

Two kinds of group-based trajectories were identified for each of three-factor variables: a long-term and a short-term improvement group. Sex and age were not associated with being classified into short-term improvement groups, but obesity was associated for all three-factor variables with relative risk ratios (RRR) varying from 1.26 (95% CI 1.02 to 1.56) to 1.45 (95% CI 1.10 to 1.90). Preoperative leg and back pain duration was significantly associated solely with back pain severity with an RRR of 1.28 (95% CI 1.01 to 1.61).

**Data Availability Statement:** The data used in this study were part of electronic patient record kept by the wellbeing services county of Southwest Finland. The data were delivered to the research

team by the IT-services as an anonymous dataset. The raw data cannot be passed on to other parties. Limited dataset used in this particular analysis can be delivered by a corresponding author on a reasonable request. Additional requests regarding the data availability can be addressed to the ethical board of the wellbeing services county of Southwest Finland eettinen.toimikunta@varha.fi.

**Funding:** The author(s) received no specific funding for this work.

**Competing interests:** The authors have declared that no competing interests exist.

## Conclusions

The results of this study suggest that most of the patients may experience pain relief and improved functioning within three months after lumbar surgery, and this effect may last, at least, for two years. Higher BMI and worse preoperative pain and disability were associated with the inferior outcome of surgery. When considering surgery, planning pre- and postoperative rehabilitation, or forecasting the use of pain medications, a higher probability of worse outcome could be expected for overweight and initially more painful patients with higher level of disability.

## Introduction

Spinal surgery is supposed to reduce disability and pain while enhancing the quality of life of a patient [1]. Despite the substantial body of knowledge concerning the average changes in pain and disability level after surgery, less is known about these changes in different subgroups. It is self-evident that, while most patients may demonstrate trajectories of these changes close to the average one, some groups may follow more or less different trends.

The group-based trajectory analysis (GBTA) is a finite mixture modelling application that has been used to distinguish groups of individuals with similar, by group, trajectories over time [2, 3]. Among other populations, this approach has previously been applied among patients undergoing lumbar spinal surgery [4–13]. Hebert et al. [10] have used the GBTA among patients undergoing surgery due to lumbar spinal stenosis. They have observed three trajectories for back and leg pain and disability. For example, three clusters were identified to describe changes in leg pain: groups with excellent (14%), good (50%), and poor outcome (36%). Similarly, three-cluster models have been observed by other previous reports employing the GBTA in similar cohorts [7, 12]. Willems et al. [13] have studied the trajectories of leg and back pain and functioning with patients undergoing lumbar microdiscectomy. The results were like the results by Hebert et al.–up to 80% of the cohort demonstrated great improvement after surgery, but there also were groups with moderate or minimal change or even a relapse after the first three months during a two-year follow-up.

Some factors have been proposed to explain the differences in the outcome of lumbar surgery. These factors might affect changes in back and leg pain as well as changes in disability severity. Previous studies have linked obesity with worse pain and disability after lumbar decompression surgery [14, 15]. The results have been inconsistent as other previous reports have suggested that obesity does not necessarily correlate with clinically significant deterioration in functioning and worse pain after surgery, even if obesity might predict elevated rates of complications [16–20]. Donnarumma et al. [21] have reported that women might benefit less from lumbar decompression surgery than men. On the other hand, multiple studies have reported that sex does not affect the magnitude of improvement after lumbar surgery in terms of functioning and quality of life, even though women might experience worse preoperative pain than men do [22–24]. Murphy et al. [25] have reported that elderly patients might increase the risk of complications and longer stay in a hospital after lumbar decompression surgery. Then again, previous studies have reported that lumbar decompression might significantly improve functioning in the long term even among patients over 80 years [26–28]. The longer duration of pre-operative pain has been connected to worse post-operative pain and greater disability [29, 30].

While the use of GBTA has increased in the past years, the need for large longitudinal analyses has been noticed by previous research [4, 10]. The evidence regarding risk factors associated with modest or poor results of surgery in terms of pain and disability is limited [10, 11]. An ability to predict who may benefit from the surgery the most and who does not improve or improve only slightly may be of help when planning surgical procedures. Identifying different postoperative trajectories of disability and pain may provide valuable insights for optimizing pre- and postoperative rehabilitation strategies.

The objective of this study was to identify groups exhibiting distinct trajectories within the broader cohort of patients undergoing lumbar spine surgery and to determine whether certain factors may be associated with the probability of being classified into a particular group. The common goal of lower back surgery is to alleviate radiating lower extremity pain in particular. This study also aimed to find out whether the change in the intensity of low back pain followed the change in lower extremity pain or whether the trajectories of these changes were different postoperatively. The registry data used in this study included both surgeries where fusion techniques were used and surgeries without fusion. It has been assumed here that the indications for surgery in these two groups might differ substantially. Therefore, changes in pain intensity were also analyzed for these two surgical subtypes as a sensitivity test.

## Materials and methods

The study cohort consisted of all consecutive patients undergoing lumbar spinal surgery of any kind in a university hospital between June 21, 2018, and August 17, 2021. Patients responded to a repeated questionnaire at four timepoints: a) $< = 2$ months before the surgery (baseline wave #1); two to four months after the surgery (wave #2); 11 to 13 months after the surgery (wave #3); and 23 to 25 months after the surgery (wave #4). Due to the retrospective study design, the timeframe (2–4, 11-13-, and 23-25-months post-surgery) for data collection has been determined by the clinical practice existed in the clinic–patients have visited the clinic at these time points for post-operative controls. The survey has been described in detail somewhere else [31]. The included procedure codes according to the Nordic Classification of Surgical Procedures (NCSP), version 1.15. are presented in Table 1. All the patients, who have undergone more than one procedure during a follow-up, were excluded.

The study was a register-based and the data were derived from an electronic patient record. By the decision of the institutional ethical board of the well-being services county of Southwest Finland, register-based studies in the institution do not require a separate statement of approval. All data used for the analysis were extracted and processed anonymously and the

**Table 1. Procedure codes.**

| Surgery code[a] | Procedure |
|---|---|
| ABC36 | Decompression of lumbar nerve roots |
| ABC56 | Decompression of lumbar spinal canal and nerve roots |
| ABC66 | Decompression of lumbar spinal channel |
| NAG61 | Posterior fusion of lumbar spine without fixation |
| NAG62 | Posterior fusion of lumbar spine with fixation, 2–3 vertebrae |
| NAG63 | Posterior fusion of lumbar spine with fixation >3 vertebrae |
| NAG66 | Posterior interbody fusion of lumbar spine, 2 vertebrae |
| NAG67 | Posterior interbody fusion of lumbar spine >2 vertebrae |

[a]Nordic Classification of Surgical Procedures NCSP; International Classification of Diseases ICD-10

need for informed consent from patients was waived by the previously mentioned ethical board. The data were accessed for research purposes on March 1st, 2024.

The Oswestry Disability Index (ODI) is a questionnaire containing 10 items covering disability caused by low back pain [32, 33]. Each item was assessed on a six-level ordinal scale with '0' describing 'no limitation' and '5' describing 'extreme limitation or an inability to function'. The total score is calculated as follows: 'Total score = (∑item scores/50) x 100'. The Finnish version of the ODI was used [34]. ODI results were interpreted as follows: 0 to 20 points: Minimal disability, 21 to 40 points: Moderate disability, 41 to 60 points: Severe disability, 61 to 80 points: Crippled, 81 to 100 points: Bed-bound [32, 33].

Back and leg pain intensity was assessed using the visual analog scale (VAS) varying from 0 to 100 points, with 0 indicating 'no pain' and 100 indicating the 'worst possible pain'. VAS results were interpreted as follows: 0 to 4 points, no pain; 5 to 44 points, mild pain; 45 to 74 points, moderate pain; and 75 to 100 points, severe pain [35].

Age was defined in full years at the time of surgery. For the main analysis, the sample was divided into four equal age groups: n = 362 (age 50.2 [8.5] years), n = 363 (age 64.6 [2.9] years), n = 363 (age 72.5 [1.8] years) and n = 363 (age 80.2 [3.2] years), respectively. In turn, when calculating relative risk ratios (RRR) by using a logistic regression model, the sample was divided into two equal age groups: n = 725 (age 57.4 [9.6] years) and n = 726 (age 76.4 [4.7] years, respectively. Correspondingly, when calculating RRRs, preoperative leg and back pain duration was dichotomized as $> = 3$ months vs. $<3$ months, and BMI was dichotomized as $> = 30$ kg/m$^2$ vs. $<30$ kg/m$^2$.

## Statistical analysis

The logical chain of GBTA analysis employed here was as follows. The first important question was how many groups (clusters) should be included in the analysis. The number of different trajectories can vary from one (the change in the average of the entire data set) to the n number of the sample (the trajectory is analyzed for each individual person). Another important choice was the shape of the regression curve (linear, quadratic, cubic, etc.). Based on these two issues, a model that would meet certain following requirements was preferred. There are no commonly accepted exact recommended selection criteria for the number of trajectories. It greatly depends on a sample size. Here the limit was set that none of the clusters should be smaller than 10% of the sample size. This cut-off was arbitrary and it was based on the sample size and the experience of the authors.

As additional confirmation, another criterion was set - in no cluster should the average posterior probability (APP) be lower than 0.7 [36]. Regarding the degree of the regression curve, a common practice is to try models of different degrees and use in the analysis the highest possible degree, which produces a statistically significant result (p<0.05).

The goodness of fit of the selected model was assured based on additional statistics: the Bayesian Information Criterion (BIC) and the Akaike Information Criterion (AIC) were calculated for the different combinations of the numbers of clusters and the regression degrees making sure that their values were closer to zero compared to models based on a smaller number of clusters or other regression degrees [37]. Trajectory analysis was repeated separately for the ODI and the severity of leg and back pain. The models chosen and their goodness of fit are presented in S1 Table.

A two-cluster model demonstrated the best fit for changes in disability severity and leg pain. While a three-cluster model demonstrated an acceptable fit for back pain, the smallest average posterior probability for this model was close to a pre-agreed cut-off of 0.7 and the size of the smallest group was close to a pre-agreed cut-off of 10%. In order to ensure the easiness

of interpretation of the results, further analysis was conducted using two-cluster models for all three-factor variables–disability, back pain, and leg pain.

The sensitivity test was performed by comparing those who underwent fusion (NAG61, NAG62, NAG63, NAG66, NAG67) vs. no-fusion (ABC36, ABC56, ABC66) surgery.

After identifying clusters, the probability of group membership was calculated based on sex, age, the leg and back pain duration before surgery, and obesity. These probabilities were expressed as RRR along with 95% confidence intervals (95% CI). All the analyses were conducted using Stata/IC Statistical Software: Release 18, College Station (StataCorp LP, TX, USA).

## Results

The preoperative surveys were completed by 1,451 patients whose mean age was 66.9 years. Of the patients, 793 (55%) were women and 658 (45%) were men. The mean BMI was 28.9 kg/m2 (Table 2). Of the patients, 567 (39%) reported pain for < = 3 months and 884 (61%) had experienced pain for >3 months before surgery. The average preoperative ODI score was 41.9 (16.9) points. The most common reasons for surgery and the most common surgical techniques are shown in the Table 2.

Two types of trajectories were identified for each of the three-factor variables (disability level and back and leg pain)–trajectories reflecting a substantial long-term improvement

**Table 2. Baseline characteristics of the study population.**

| Demographics, pain severity, ODI [a] score | | |
|---|---|---|
| | **Mean** | **Standard deviation** |
| Age (entire sample), years | 66.9 | 12.1 |
| Age group 1 (n = 362), years | 50.2 | 8.5 |
| Age group 2 (n = 363), years | 64.6 | 2.9 |
| Age group 3 (n = 363), years | 72.5 | 1.8 |
| Age group 4 (n = 363), years | 80.2 | 3.2 |
| Body mass index, kg/m$^2$ | 28.9 | 4.9 |
| Back pain intensity, points | 58.9 | 26.8 |
| Leg pain intensity, points | 63.6 | 26.3 |
| ODI [a] total score, points | 41.9 | 16.9 |
| Pain duration before surgery | N | % |
| <3 months | 567 | 39 |
| > = 3 months | 884 | 61 |
| Surgery codes [b] | N | % |
| ABC36 Decompression of lumbar nerve roots | 418 | 29 |
| ABC56 Decompression of lumbar spinal canal and nerve roots | 412 | 28 |
| NAG62 Posterior fusion of lumbar spine with fixation, 2–3 vertebraea | 370 | 26 |
| Others | 251 | 17 |
| Main diagnoses [c] | N | % |
| M48 Spondylopathies | 862 | 59 |
| M43 Deforming dorsopathies | 224 | 15 |
| G55/M51 intervertebral disc disorders | 144 | 10 |
| M47 Spondylosis | 114 | 8 |
| Others | 107 | 7 |

[a] Oswestry Disability Index
[b] Nordic Classification of Surgical Procedures NCSP; International Classification of Diseases ICD-10

(**long-term improvement group**) and trajectories showing only a mild short-term improvement (**short-term improvement group**). The ODI scores and leg and back pain severity at different waves by trajectory groups have been presented in S2 Table.

### Changes in disability level (Fig 1)

Of the patients, 83% were classified into a long-term improvement group. In this group, the initial average disability level was moderate 36.7 (95% CI 35.5 to 37.9) points. At three months after surgery, the disability level decreased to minimal 15.0 (95% CI 13.0 to 17.1) points and stayed around this level until the end of a two-year follow-up.

The smaller short-term improvement group included those 17% of patients, who had initially experienced severe average disability of 57.6 (95% CI 55.2 to 60.0) points. While some improvement was seen in three months after surgery, disability remained severe from 41.6 (95% CI 38.1 to 45.0) to 46.5 (95% CI 42.7 to 50.3) points, showing some worsening towards the end of the two-year follow-up.

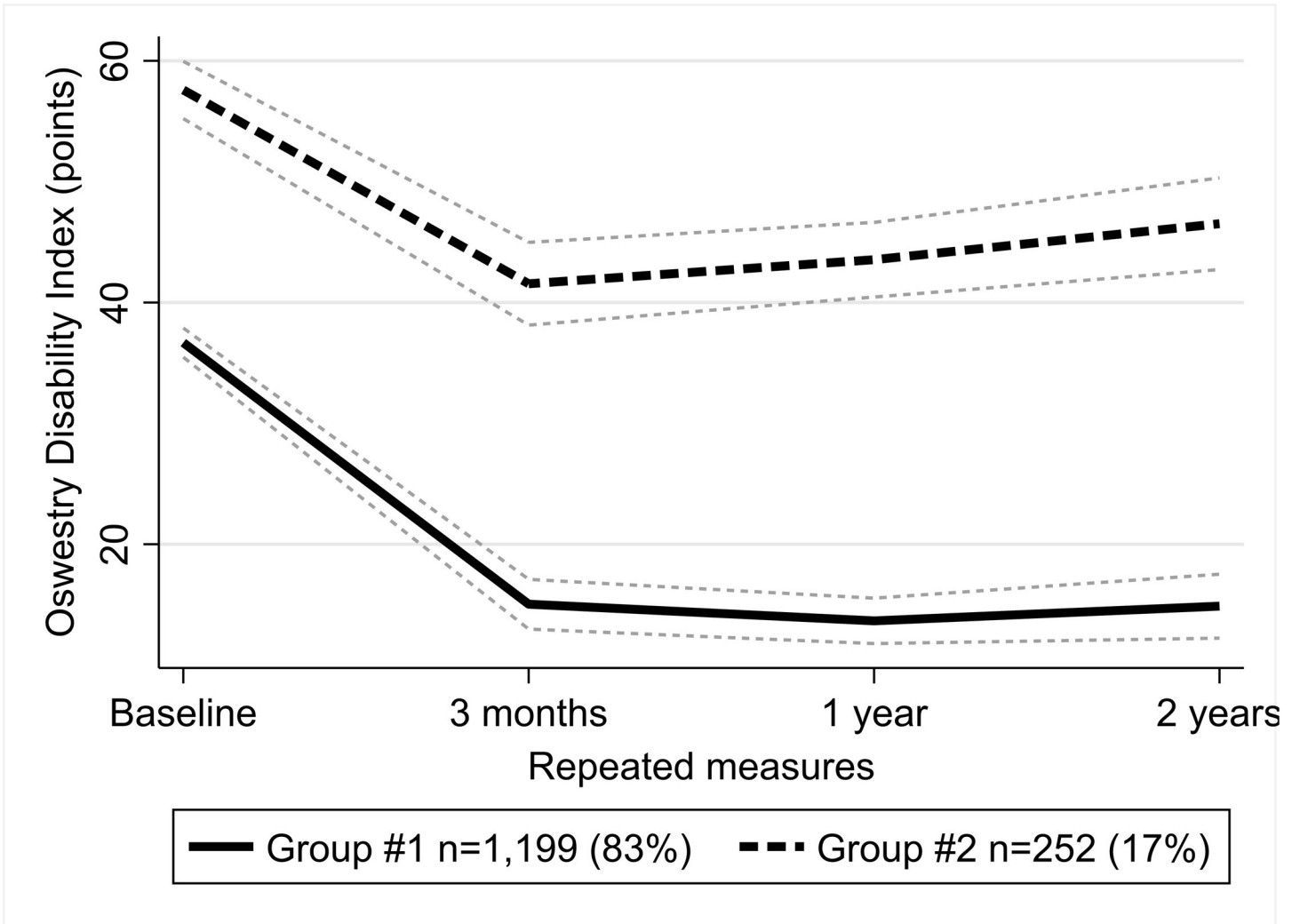

**Fig 1. Trajectories of the ODI.** Group #1 – long-term improvement group; group #2 – short-term improvement group.

### Changes in back pain severity (Fig 2)

Of the patients, 69% were classified into a long-term improvement group. In this group, the initial average pain level was moderate at 51.9 (95% CI 47.9 to 55.9) points. At three months after surgery, the pain level decreased down to mild 16.1 (95% CI 12.8 to 19.3) points and stayed this level until the end of the two-year follow-up.

The smaller short-term improvement group included those 31% of patients, who had initially experienced moderate pain of 68.0 (95% CI 62.8 to 73.2) points. While some improvement was seen three months after surgery, the pain worsened after that, until reaching the level of 66.0 (95% CI 59.4 to 72.6) points at the end of a two-year follow-up, the estimate close to a baseline level when taking into account its 95% CIs.

### Changes in leg pain severity (Fig 3)

Of the patients, 63% were classified into a long-term improvement group. In this group, the initial average pain level was moderate at 57.7 (95% CI 53.2 to 62.2) points. At three months

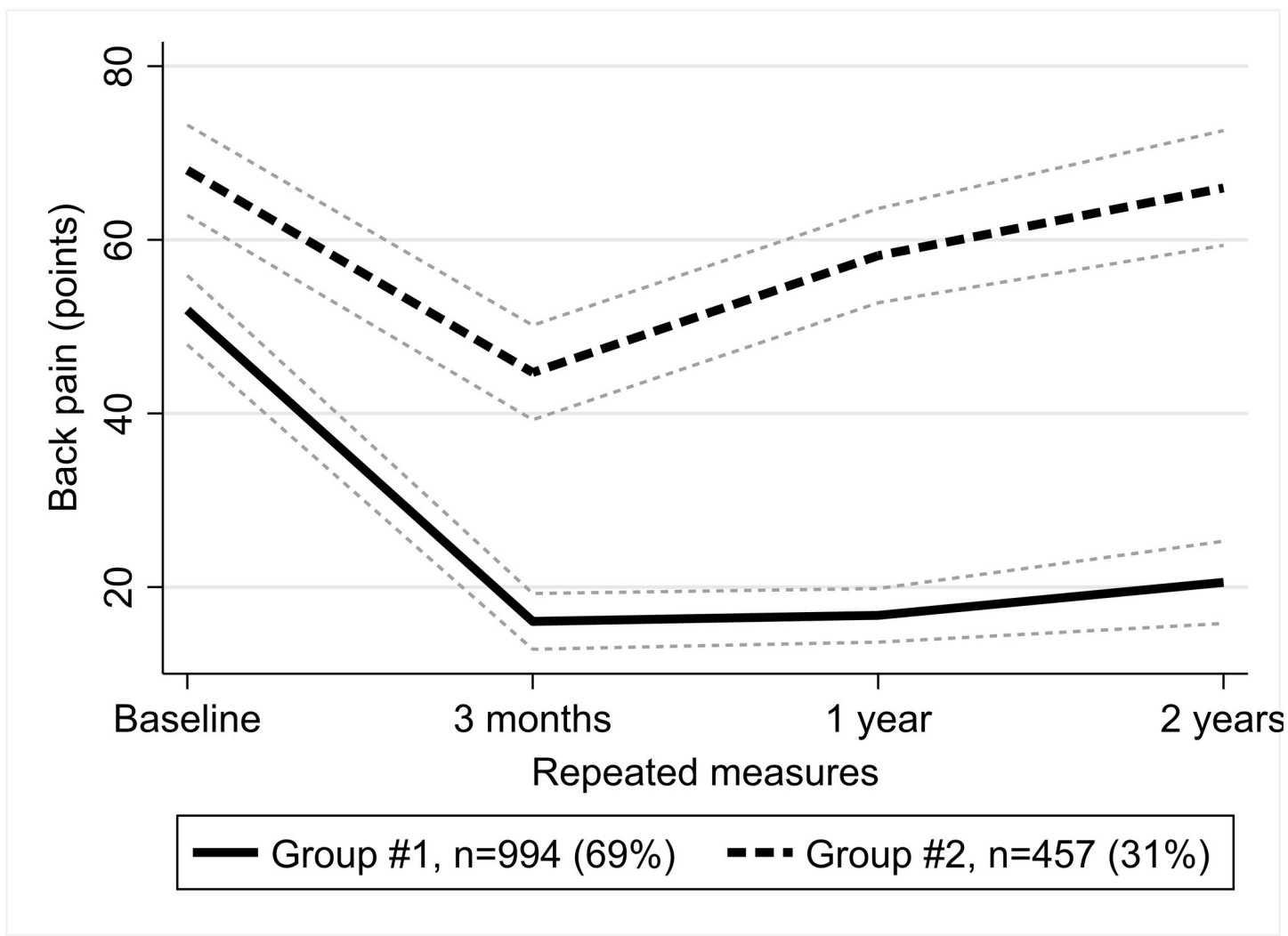

**Fig 2. Trajectories of back pain.** Group #1 – long-term improvement group; group #2 – short-term improvement group.

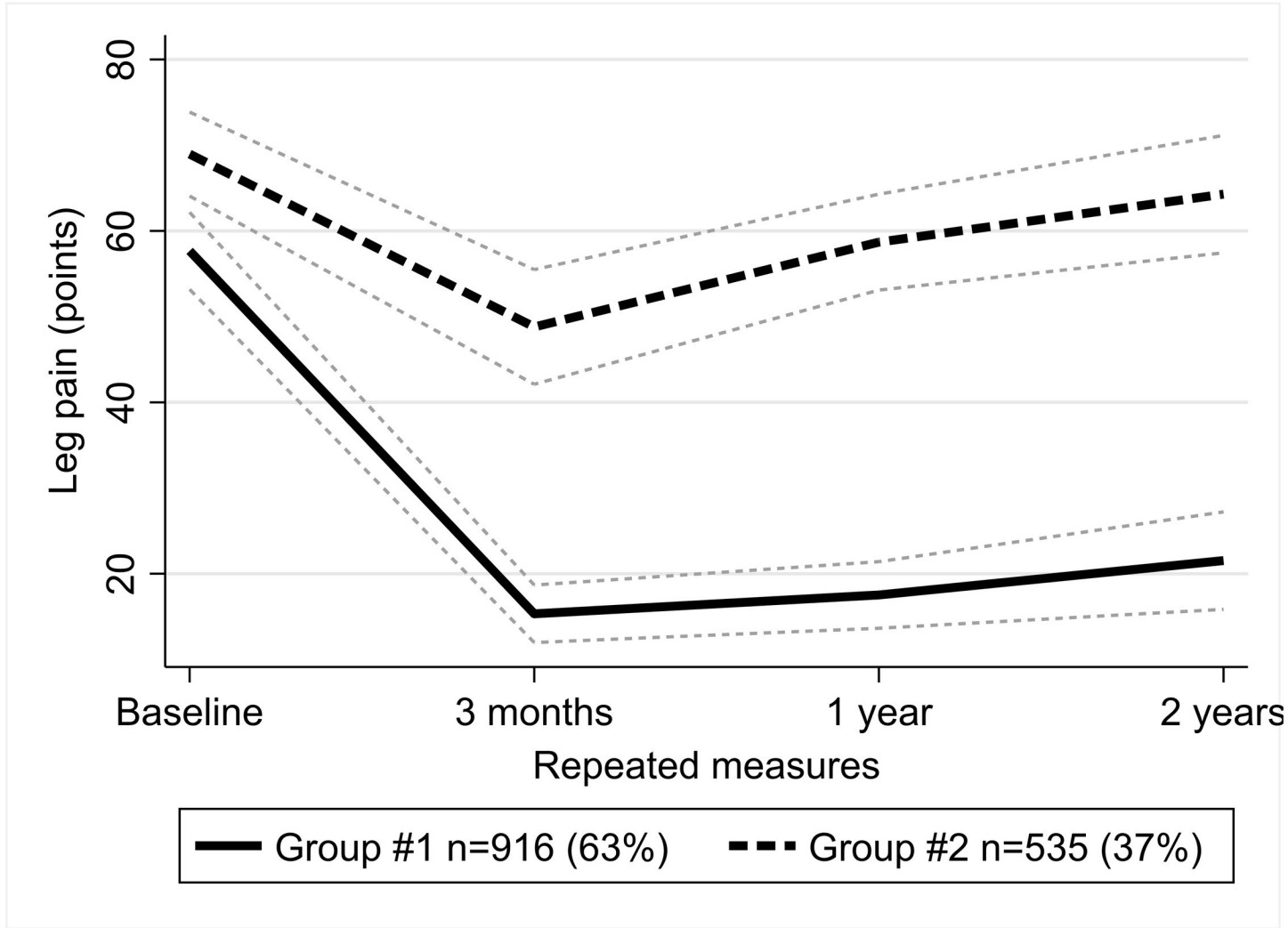

**Fig 3. Trajectories of leg pain.** Group #1 – long-term improvement group; group #2 – short-term improvement group.

after surgery, the pain level decreased down to mild 15.3 (95% CI 12.0 to 18.7) points and stayed around this level until the end of the two-year follow-up.

The smaller short-term improvement group included those 37% of patients, who had initially experienced moderate average pain of 69.0 (95% CI 64.1 to 73.9) points. While some improvement was seen three months after surgery, the pain worsened after that, until it reached the level of 64.3 (95% CI 57.5 to 71.1) points at the end of a two-year follow-up, the estimate close to a baseline level when taking into account its 95% CIs.

### Association between demographic factors and probability of being classified into a short-term improvement group

Sex and age were not significantly associated with a risk of being classified in a short-term improvement group for any of the three-factor variables (Table 3). Preoperative duration of pain was significantly associated with that probability for back pain severity showing the RRR of 1.28 (95% CI 1.01 to 1.61) which was close to the level of non-significance. Instead, BMI$\geq$30 kg/m$^2$ was connected to a probability of being classified into of short-term improvement

**Table 3. Relative risk ratios (RRRs) of being classified to a short-term improvement group.** Reference is long-term improvement group. RRRs are adjusted for age and sex.

| Variable/group | RRR | 95% CI | |
|---|---|---|---|
| Groups based on disability level | | | |
| Women vs. men | 1.00 | 0.76 | 1.32 |
| Pain duration > = 3 months vs. <3 months | 1.09 | 0.82 | 1.45 |
| BMI ≥30 vs. BMI <30 | 1.45 | 1.10 | 1.90 |
| Age (older vs younger) | 1.06 | 0.81 | 1.39 |
| Groups based on back pain severity | | | |
| Women vs. men | 1.09 | 0.87 | 1.37 |
| Pain duration > = 3 months vs. <3 months | 1.28 | 1.01 | 1.61 |
| BMI ≥30 vs. BMI <30 | 1.32 | 1.05 | 1.65 |
| Age (older vs younger) | 0.86 | 0.69 | 1.07 |
| Groups based on leg pain severity | | | |
| Women vs. men | 1.17 | 0.94 | 1.45 |
| Pain duration > = 3 months vs. <3 months | 1.05 | 0.84 | 1.31 |
| BMI ≥30 vs. BMI <30 | 1.26 | 1.02 | 1.56 |
| Age (older vs younger) | 1.13 | 0.92 | 1.40 |

group for all three factor variables (disability and back and leg pain) with RRR varying from 1.26 (95% CI 1.02 to 1.56) to 1.45 (95% CI 1.10 to 1.90).

## Fusion vs. No-fusion

The sensitivity analysis (fusion vs. no-fusion) showed trajectories that were similar to the trajectories seen for the entire sample–larger groups demonstrated long-term improvement, while smaller groups did not (S1 Fig). There were, however, some differences between descriptive baseline characteristics in fusion and no-fusion groups (S3 Table). As for the entire sample, overweight was the most common significant predictor of being classified to a group with only short-term improvement (S4 Table).

## Discussion

This observational registry-based study of nearly 1,500 patients undergoing lumbar spine surgery identified different trajectories of change in pain and disability severity in a two-year follow-up after surgery. These trajectories described either long-term or short-term improvements. Most of the patients (up to four-fifths) could be classified into groups with long-term improvement showing substantial positive trends in both disability and pain during the entire follow-up. The rest of the patients showed only temporary short-term improvement, which diminished after three postoperative months. The probability of being classified into groups with short-term improvement was mostly associated with worse pain and disability before surgery and higher BMI. Sex and age were not significantly associated with the risk of being classified into short-term improvement groups. For back pain, this probability was loosely associated with the longer duration of preoperative pain.

The present results are in line with some previous research. For example, Li et al. have identified two trajectory groups in a population of patients undergoing spinal surgery: a larger group showing mild post-operative pain and a smaller group showing relapsing pain [4]. Some previous studies have observed three or even four distinguishable trajectory groups among populations that were similar to the cohort studied here [6, 7, 10, 12, 13]. For example, Willems et al. [13] have observed four different trajectories of change in the severity of leg pain among

patients undergoing microdiscectomy: groups with large (79%), moderate (8%), and minimal improvement (5%) and groups with pain relapse (8%). Differences between studies in the number of trajectories might be related to purely medical heterogeneity (diverse indications for surgery or different surgery techniques), as well as to the dissimilar sample sizes, settings, number of repeated measures, the duration of follow-up, distributions of demographic characteristics, etc. Also, some studies have employed outcome measures other than VAS or ODI [7, 10, 12]. It should also be taken into account that setting cut-offs for the number of trajectories detected through the GBTA is arbitrary and often based on logic and common sense rather than on exact mathematics. Despite these differences, the common trend could be seen in both the present and previous reports–while most of the patients demonstrate substantial improvements in pain and disability after spine surgery, there is a smaller group, that either improve only for a little while or does not improve at all. The momentary improvement of pain and functioning in the short-term improvement groups could be explained by placebo effect. In 2021, Jamjoom et al. [38] reported in a systematic review that the placebo effect could last up to six months after spinal surgery and it can be seen in patient-reported outcome measures (PROMs). Other explanations for this momentary improvement and then the return of worse pain or functioning could be increased pain medication post-operatively or the development of a clinical complication such as infection or failure of fixation over time. It is also possible that for some of the patients, functioning may be poor for some reason other than the condition being treated, and the surgery aims for example to reduce radicular pain.

In agreement with the present findings, obesity has previously been associated with worse surgical outcome [14, 15]. Park et al. [15] have observed the association between BMI≥30 and inferior surgical results in a cohort of 32,000 patients undergoing lumbar spinal surgery. The evidence of such a relationship has, however, been controversial. In their systematic review, Ghobrial et al. [20] have stated that obesity is not related to higher post-operative VAS-measured back pain, leg pain, or functioning measured with ODI in populations undergoing non-instrumented lumbar spinal surgery according to minimal clinical important difference (MCID) [20]. However, in that review by Ghobrial et al, it was found that although overweight subjects had a significant improvement according to the MCID value, they still had greater pain both at baseline and post-operatively in almost all studies. Some other studies have also stated that, even though obesity is associated with more postoperative complications, even overweight people may benefit from surgery [16, 18, 19]. However, Onyekwelu et al. [19] reported that although overweight people benefited from surgery, normal weight people benefited more. Additionally, Lingutla et al. [18] reported that problems arose in their study due to the non-standard definition of obesity and the limited availability of data. Also, the study by Divi et al. [2] had a small sample size. Even more, Rihn et al. [17] reported that obese patients may benefit more from surgery compared to nonoperative treatment than normal weight patients. On the other hand, this result could be explained by the fact that the response of overweight patients to nonoperative treatment was poor. In 2016, Jackson et al. [39] published a systematic review where the effects of obesity on spine surgery were discussed. In that review, it was reported that overweight people have a higher incidence of back pain, radiating pain and a higher rate of disk degeneration. In the review, possible causes of increased disc degeneration were found, for example, to be greater mechanical compression on the spine, chronic inflammation related to obesity and reduced blood flow to the disc. Being overweight has also been linked to a higher risk of complications in previous studies [16–20].

In this study, neither gender nor age increased the probability of being categorized into groups with a worse outcome. Similar results have also been obtained from previous studies [22–24, 26–28]. Donnarumma et al. [21] reported that women may benefit less from lumbar decompression surgery. However, their study was conducted with a small sample size.

According to the study by Murphy et al. [25] older age increases the risk of complications, longer hospital stay, and discharge to a place other than home. However, their study did not address disability or pain.

Higher and prolonged pre-operative pain and disability have also previously been found to affect their developmental trajectories after surgery [6, 7, 13, 40]. Carrasco et al. [7] have reported that worse pre-operative low back pain might be a risk of being categorized into groups with moderate or no improvement after lumbar decompression and microdiscectomy. Also, worse preoperative leg pain was associated with a higher probability of being classified into a group of no-improvement. Prolonged pain maintenance even after the surgery could be explained by central sensitization. Prolonged chronic pain may result in functional, chemical and even structural changes in central nervous system. If so, then nervous system can be persistently in a high activity state and a person can experience pain even in the absence of nociceptive input, which has successfully been treated with surgery [41].

The generalizability of these findings might be limited due to several issues. The study was set in a highly specified university surgery clinic, and the results might be different in primary care or non-university clinics. Even though most of the patients were undergoing spinal decompression and fusion, the indications for surgery were rather heterogeneous. The data included information from patients who have undergone only one back surgery. However, this restriction only applied to data after the start of registry data collection. It is therefore possible that there were individual patients who underwent more than one operation, if the operations had been performed before the registry data started to be collected. This is a commonly known weakness of register data, a truncation at the beginning of follow-up, which complicates the differentiation between prevalent and incident cases. This weakness could increase the heterogeneity of the sample. As patients with multiple back surgeries have been excluded, the characteristics of the studied sample could deviate from the real-life situation, when patients with degenerative spine diseases often have to undergo several operations. Only a few explanatory variables were available. Thus, it is possible, that some important factors affecting pain and disability after surgery remained unobserved. For example, there was no information on how many patients had diabetes mellitus or how many were smoking. Complications were not taken into account in this material, so based on the results it cannot be stated, for example, that complications caused by obesity would have worsened the treatment outcome. Register-based data commonly suffer from truncation at the start of follow-up making it difficult to differentiate between prevalent and incident cases. Such data may also be affected by unrecognized confounding information and potential data dredging [42]. The present retrospective study lacked the possibility to deal with missing data as these had not been available for analysis. Many patients did not come to the control visits at the clinic. It is uncertain if the data were missing due to random chance or due to specific reasons, e.g., people with the greatest improvement have not appeared at their post-operative control appointments or the other way round–people dissatisfied with the results of surgery were not willing participate into follow-up. It has previously been suggested that missing data can substantially affect the precision of estimated change in PROMs scores from clinical registry data [43]. The study did not explore other potentially relevant PROMs that could provide a more comprehensive assessment of surgical success (e.g., improved quality of life or return to work). This limitation could create uncertainty concerning the overall improvement after surgery. It has been previously reported that back pain and disability may correlate in some situations only loosely [44]. Thus, persistent pain does not necessarily mean unchanged quality of life, perceived general health or ability to work. Additionally, the lack of a control group made it impossible to draw any reliable inferences concerning causality. Performing a manual chart review could probably add value

to the study by revealing patient characteristics, pathologies, and surgical details, which were not available through the register.

Further research may replicate these findings in different settings, and populations and using different pre-operative independent risk factors like smoking, educational level, depression severity, etc. Especially important could be confirming the findings in groups, which are more homogenous regarding their indications for surgery and surgery techniques.

## Conclusions

The results of this study suggest that most of the patients might experience pain relief and improved functioning within three months after lumbar surgery, and this effect may last, at least, two years. However, there is a smaller group who might experience only temporary improvement, which diminishes after three months since surgery. Age, sex, and the duration of preoperative pain did not affect the risk of being classified in a particular group. Instead, higher BMI and worse preoperative pain and disability were associated with the inferior outcome of surgery. When considering surgery, planning pre- and postoperative rehabilitation, or forecasting the use of pain medications, a higher probability of worse outcomes could be expected for overweight and initially more painful patients with higher level of disability.

## Supporting information

**S1 Fig. Trajectories of changes in pain and disability after surgery comparing fusion and no-fusion techniques.**
(DOCX)

**S1 Table. Goodness of fit of group-based trajectory analysis models.**
(DOCX)

**S2 Table. ODI scores and pain severity at different waves by trajectory groups.**
(DOCX)

**S3 Table. Descriptive characteristics of sample by fusion vs. no-fusion techniques.**
(DOCX)

**S4 Table. Relative risk ratios (RRRs) of being classified to a short-term improvement group.**
(DOCX)

## Author Contributions

**Conceptualization:** Konsta Koivunen, Jari Arokoski, Sara Widbom-Kolhanen, Katri Pernaa, Juhani Juhola, Mikhail Saltychev.

**Data curation:** Katri Pernaa, Mikhail Saltychev.

**Formal analysis:** Konsta Koivunen, Jari Arokoski, Sara Widbom-Kolhanen, Katri Pernaa, Juhani Juhola, Mikhail Saltychev.

**Investigation:** Konsta Koivunen, Jari Arokoski, Sara Widbom-Kolhanen, Katri Pernaa, Juhani Juhola, Mikhail Saltychev.

**Methodology:** Konsta Koivunen, Jari Arokoski, Sara Widbom-Kolhanen, Katri Pernaa, Juhani Juhola, Mikhail Saltychev.

**Software:** Mikhail Saltychev.

**Supervision:** Mikhail Saltychev.

**Validation:** Konsta Koivunen, Jari Arokoski, Sara Widbom-Kolhanen, Katri Pernaa, Juhani Juhola, Mikhail Saltychev.

**Visualization:** Konsta Koivunen, Sara Widbom-Kolhanen, Katri Pernaa, Juhani Juhola, Mikhail Saltychev.

**Writing – original draft:** Konsta Koivunen.

**Writing – review & editing:** Jari Arokoski, Sara Widbom-Kolhanen, Katri Pernaa, Juhani Juhola, Mikhail Saltychev.

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
