## [Decision Letter · Decision Letter 0]

18 Jul 2024

PONE-D-24-24037Disability and pain after lumbar surgery – group-based trajectory analysisPLOS ONE

Dear Dr. Koivunen,

Thank you for submitting your manuscript to PLOS ONE. After careful consideration, we feel that it has merit but does not fully meet PLOS ONE’s publication criteria as it currently stands. Therefore, we invite you to submit a revised version of the manuscript that addresses the points raised during the review process.

We look forward to receiving your revised manuscript.

Kind regards,

Shabnam ShahAli, Ph.D.

Academic Editor

PLOS ONE

Journal Requirements:

2. In the online submission form you indicate that your data is not available for proprietary reasons and have provided a contact point for accessing this data. Please note that your current contact point is a co-author on this manuscript. According to our Data Policy, the contact point must not be an author on the manuscript and must be an institutional contact, ideally not an individual. Please revise your data statement to a non-author institutional point of contact, such as a data access or ethics committee, and send this to us via return email. Please also include contact information for the third party organization, and please include the full citation of where the data can be found

Reviewers' comments:

Reviewer's Responses to Questions

**Comments to the Author**

1. Is the manuscript technically sound, and do the data support the conclusions?

Reviewer #1: Yes

Reviewer #2: Yes

Reviewer #3: Yes

2. Has the statistical analysis been performed appropriately and rigorously? 

Reviewer #1: No

Reviewer #2: Yes

Reviewer #3: Yes

3. Have the authors made all data underlying the findings in their manuscript fully available?

Reviewer #1: No

Reviewer #2: Yes

Reviewer #3: No

4. Is the manuscript presented in an intelligible fashion and written in standard English?

Reviewer #1: Yes

Reviewer #2: Yes

Reviewer #3: No

5. Review Comments to the Author

**Reviewer #1:** The authors attempted to apply group-based trajectory analysis to a cohort of patients undergoing lumbar spine surgery. The topic is so interesting for rehabilitation issues, and the study was well done, that their efforts are commendable:

General comments:

1- What is your logic for analyzing and categorizing patients according to their leg pain, back pain, etc.? Is it based on the literature or did you select them exploratively? Please clarify this in the objectives of your study and also in the method.

2- Most of the study's supporting data are necessary to understand the study's results. It's better to provide them in the main text or as a figure or table.

Specific comments:

1- Abstract, p2, line 30: What does RRR stand for?

2- Introduction, p6: Please explain the objectives of the study in more detail with the rationales.

3- Material and methods, p7, line 115: Change 'logit regression model' to 'logistic regression model'.

4- Please rewrite the statistical analysis section of the manuscript. It is not clearly defined and raises so many questions for a reader.

5- Statistical analysis, p8, line 127: Why did you set the cut-off value to 10%?

6- Discussion, p15, line 241: What does PROMs stand for?

7- Discussion, p17, line 257: Please explain the probable reasons for these controversies?

8- Discussion, p17, line 267-268: What do you mean by “The register did not contain data on spinal surgery prior to the follow-up”?

9- Please define groups in Figure1, 2 &3.

**Reviewer #2:** Thank you for the opportunity to review this manuscript. There are some specific comments:

- Through the whole text please insert space between the statement and the reference. Also please insert the reference just after the author's name when it is applicable (for example line 54 after the name Willems et al.)

-please insert reference for the statement in line 101 and 102. Please do the same for lines 125 and 126.

Discussion:

-I would suggest explaining central sensitization as an explanation of pain maintenance even after the surgery (line 263).

**Reviewer #3:** The manuscript presents a well-structured observational study involving nearly 1,500 patients undergoing lumbar spine surgery, focusing on long-term outcomes such as pain relief and functional improvement over a two-year period. However, several points warrant attention:

Abstract: The background section lacks clarity and cohesion, requiring improvement. Additionally, the English writing throughout needs clarity and coherence.

Introduction: While the study's aim is clear, the background section could benefit from better cohesion leading toward the final conclusions.

Methodology: Clarification is needed regarding the administration of questionnaires to patients. The rationale behind the wide timeframe (2-4, 11-13, and 23-25 months post-surgery) for data collection should be elaborated upon. Although previous data were utilized, the specific rationale for this timeframe should be explicitly stated. Furthermore, details on how missing data were handled are crucial for assessing the study's robustness.

The study did not explore other patient-reported outcome measures (PROMs) that could provide a more comprehensive assessment of surgical success (e.g., quality of life measures, return to work). This omission and its implications should be discussed in the Discussion section.

Discussion: While the manuscript briefly compares findings with previous studies, a more comprehensive discussion integrating conflicting results and methodological differences across studies would enhance the interpretation of the findings.

The manuscript lacks exploration into potential mechanisms underlying observed associations (e.g., why higher BMI correlates with poorer outcomes). This mechanistic insight should be addressed in the discussion.

6. PLOS authors have the option to publish the peer review history of their article (what does this mean?). If published, this will include your full peer review and any attached files.

Reviewer #1: No

Reviewer #2: **Yes: **Ghazal Kharaji

Reviewer #3: **Yes: **Sahar Boozari

---

## [Author Response · Author response to Decision Letter 0]

10 Sep 2024

Professor Shabnam ShahAli, Ph.D.

Academic Editor

PLOS ONE

Dear Professor ShahAli

Thank you for the opportunity to revise our manuscript PONE-D-24-24037 entitled "Disability and pain after lumbar surgery – group-based trajectory analysis”. We greatly appreciate the time and effort you and the reviewers have dedicated to improving our work. Please find submitted the revised manuscript and a detailed, point-by-point response to the reviewers' comments.

On behalf of all the authors

Konsta Koivunen

---

## [Decision Letter · Decision Letter 1]

28 Oct 2024

Disability and pain after lumbar surgery – group-based trajectory analysis

PONE-D-24-24037R1

Dear Dr. Koivunen,

We’re pleased to inform you that your manuscript has been judged scientifically suitable for publication and will be formally accepted for publication once it meets all outstanding technical requirements.

Kind regards,

Shabnam ShahAli, Ph.D.

Academic Editor

PLOS ONE

Additional Editor Comments (optional):

Reviewers' comments:

Reviewer's Responses to Questions

**Comments to the Author**

1. If the authors have adequately addressed your comments raised in a previous round of review and you feel that this manuscript is now acceptable for publication, you may indicate that here to bypass the “Comments to the Author” section, enter your conflict of interest statement in the “Confidential to Editor” section, and submit your "Accept" recommendation.

Reviewer #1: (No Response)

Reviewer #2: All comments have been addressed

Reviewer #3: All comments have been addressed

2. Is the manuscript technically sound, and do the data support the conclusions?

Reviewer #1: (No Response)

Reviewer #2: Yes

Reviewer #3: Yes

3. Has the statistical analysis been performed appropriately and rigorously? 

Reviewer #1: (No Response)

Reviewer #2: Yes

Reviewer #3: Yes

4. Have the authors made all data underlying the findings in their manuscript fully available?

Reviewer #1: (No Response)

Reviewer #2: Yes

Reviewer #3: Yes

5. Is the manuscript presented in an intelligible fashion and written in standard English?

Reviewer #1: (No Response)

Reviewer #2: Yes

Reviewer #3: Yes

6. Review Comments to the Author

Reviewer #1: (No Response)

Reviewer #2: all comments have been addressed. please include less figures and tables in the manuscript if it is possible.

Reviewer #3: (No Response)

7. PLOS authors have the option to publish the peer review history of their article (what does this mean?). If published, this will include your full peer review and any attached files.

Reviewer #1: No

Reviewer #2: No

Reviewer #3: **Yes: **Sahar Boozari

---

## [Editor Report · Acceptance letter]

30 Oct 2024

PONE-D-24-24037R1 

PLOS ONE

Dear Dr. Koivunen, 

I'm pleased to inform you that your manuscript has been deemed suitable for publication in PLOS ONE. Congratulations! Your manuscript is now being handed over to our production team.

Kind regards, 

on behalf of

Dr. Shabnam ShahAli 

Academic Editor

PLOS ONE